# The emergence of visual simulation in task-optimized recurrent neural networks

## Abstract

Primates display remarkable prowess in making rapid visual inferences even when sensory inputs are impoverished. One hypothesis about how they accomplish this is through a process called visual simulation, in which they imagine future states of their environment using a constructed mental model. Though a growing body of behavioral findings, in both humans and non-human primates, provides credence to this hypothesis, the computational mechanisms underlying this ability remain poorly understood. In this study, we probe the capability of feedforward and recurrent neural network models to solve the *Planko* task, parameterized to systematically control task variability. We demonstrate that visual simulation emerges as the optimal computational strategy in deep neural networks only when task variability is high. Moreover, we provide some of the first evidence that information about imaginary future states can be decoded from the model latent representations, despite no explicit supervision. Taken together, our work suggests that the optimality of visual simulation is task-specific and provides a framework to test its mechanistic basis.

## 1   Introduction

A longstanding goal in the brain sciences is to understand the neural algorithms and computations that support humans' ability to interact optimally with their surroundings. A popular cognitive level theory for how humans visually reason about their environments, under uncertainty, is that they rely on "simulation" through rich internal generative models of the world Kersten & Yuille (1996); Tenenbaum et al. (2011); Battaglia et al. (2013); Ullman et al. (2017) to build and test hypotheses about the future and plan effective behavior. The notion of visual simulation has been discussed since at least Descartes, who theorized that this ability is implemented in the brain through the same neural mechanisms as perception, and operates without any stimulation from the external world Lokhorst (2005). Ullman expanded upon this theory in his seminal Visual Routines (Ullman (1984)), in which he suggested that in order for visual simulation to work effectively it must utilize syntactic computations, which can be flexibly re-applied to any visual features. Recent studies in humans and non-human primates have provided insight into the potential neural underpinnings of these cognitive-level theories Ahuja et al. (2022); Ahuja & Sheinberg (2019); Rajalingham et al. (2021, 2022). While Ahuja & Sheinberg (2019) demonstrated the ability of simple feedforward neural networks (FFNs) to perform their visual simulation task, they find a misalignment between model and primate behavior. Similarly, Rajalingham et al. (2021) show this misalignment in recurrent neural networks (RNNs) trained to play a simplified version of Pong (*M-Pong*) where the RNN had to guess where to move a paddle to catch a linearly-moving ball. However, the authors found that the same RNNs, when trained to predict the position of M-Pong balls across their trajectories, were

Submitted to 4th Workshop on Shared Visual Representations in Human and Machine Visual Intelligence (SVRHM) at NeurIPS 2022. Do not distribute.

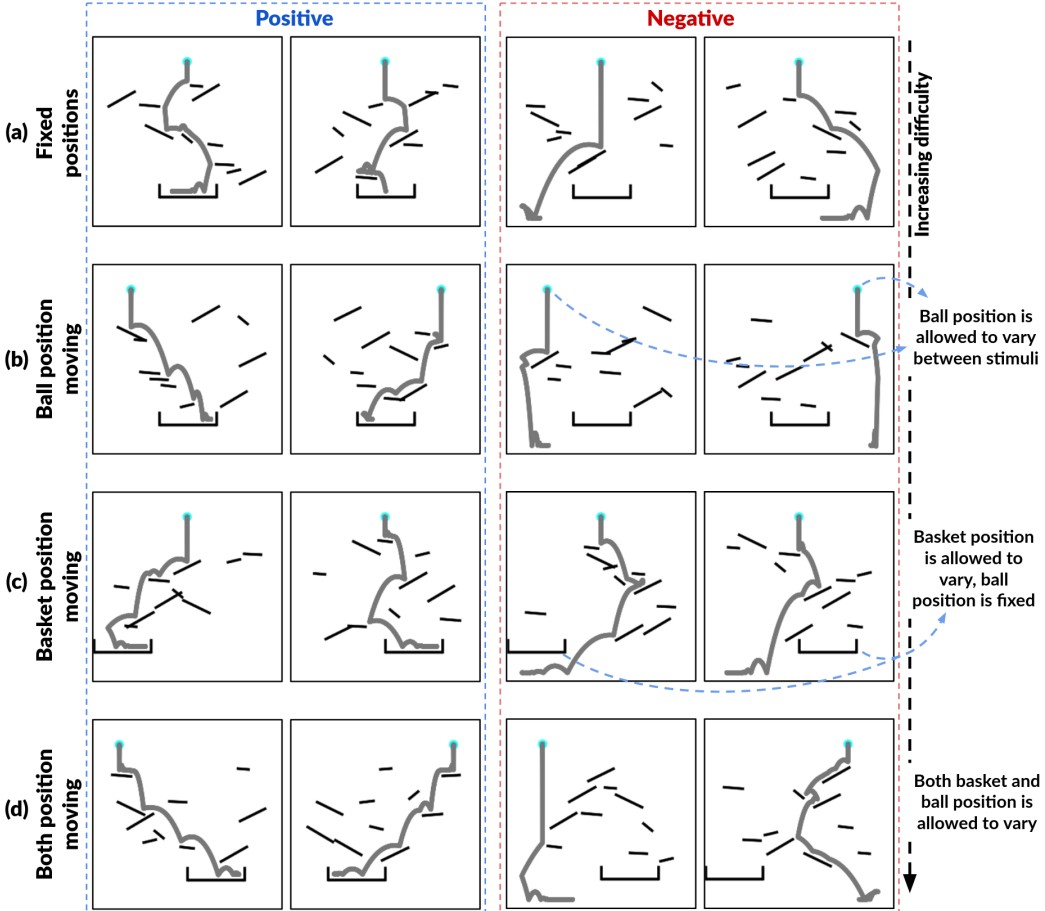

Figure 1: *Planko* **as a visual simulation task**. Observers are shown a single image of a game board and asked to predict whether the ball at the top of the image (in light blue) will fall into the catcher at the bottom of the image. The grey lines in each image depict the trajectory of the ball at the top, simulated in a world with Newtonian physics. We generate positive and negative game boards for every ball and cup position, in which the planks are placed in a way to bounce the ball into the catcher or not. We also generate four different versions of the game, in which the properties of the ball and cup are modified: (*a*) Fixed positions of each, (*b*) the ball position is randomly sampled, (*c*) the bucket position is randomly sampled, and (*c*) both the ball and bucket positions are randomly sampled. Each modification increases the total number of game boards that can be generated, and hence, the game's difficulty.

able to learn routines for visual simulation that explained significantly more variance in behavior and neural activity than RNNs without this constraint.

**Contributions** We explore the conditions under which visual simulation naturally emerges as the optimal computational strategy in deep neural networks purely driven by task-constraints. We refrain from providing any source of information about temporal dynamics to our network models, for example, watching object trajectories or explicit supervision about the locations of objects in the world. We start with the visual simulation task developed by Ahuja & Sheinberg (2019) and adapt it to our suite of models and call it *Planko* (Figure. 1). In *Planko*, observers are tasked with predicting the outcome of a ball falling through a random series of oriented planks without ever seeing the ball's trajectory. Unlike *M-Pong*, *Planko* is parameterized to make it possible to generate game boards that range from trivially easy to extremely difficult. We investigate whether models can learn to solve *Planko*, and whether the solutions they learn resemble the Newtonian physics used to generate game boards despite having no explicit access to that information.

- We find that a variety of feedforward deep neural networks (FFNs) and RNNs learn accurate solutions for easy versions of *Planko*, but only an attentional circuit model grounded in neurobiology can solve harder versions of the task (InT, Linsley et al. 2021).

- The InT's attention maps indicate that it learns to focus on paddles that may interact with the *Planko* ball, and regions of the game board where it expects the ball to fall through.

- A decoding analysis demonstrated that the InT incrementally simulates a *Planko* ball's trajectory through the game board, in hard but not easy game boards, and that this path closely approximates the ground-truth trajectory generated in each board with Newtonian physics.

- Our findings indicate that robust visual simulation emerges as an optimal algorithm in difficult environments, and that prior work Rajalingham et al. (2021, 2022) suggesting that additional learning constraints are needed for visual simulation may be a byproduct of a trivial task driving models to learn shortcuts.

## 2   The *Planko* challenge

The *Planko* challenge is inspired by prior work in visual simulation, which measured primate accuracy in simulating the trajectory of moving balls, and used fMRI to identify regions of cortex that correlated with their behavior (Ahuja et al., 2022). Much like *M-Pong*, models trained on that task learned shortcut solutions to solve it (Ahuja & Sheinberg, 2019). With our *Planko* challenge, we have controlled for variations in the task space to explicitly prevent the learning of shortcut solutions, and in order to understand the extent to which it changes the strategies learned by models for visual simulation.

Each *Planko* board depicts a ball at the top of the screen placed above ten randomly oriented and positioned planks. A catcher is placed at the bottom of the screen (Figure. 1). Each plank is parameterized by its angle of inclination, length, and its position on the screen. The *Planko* ball and catcher are placed in accordance with task difficulty as discussed below. The physics of this world are specified by Newton Dynamics (`http://www.newtondynamics.com`). The ball's trajectory as

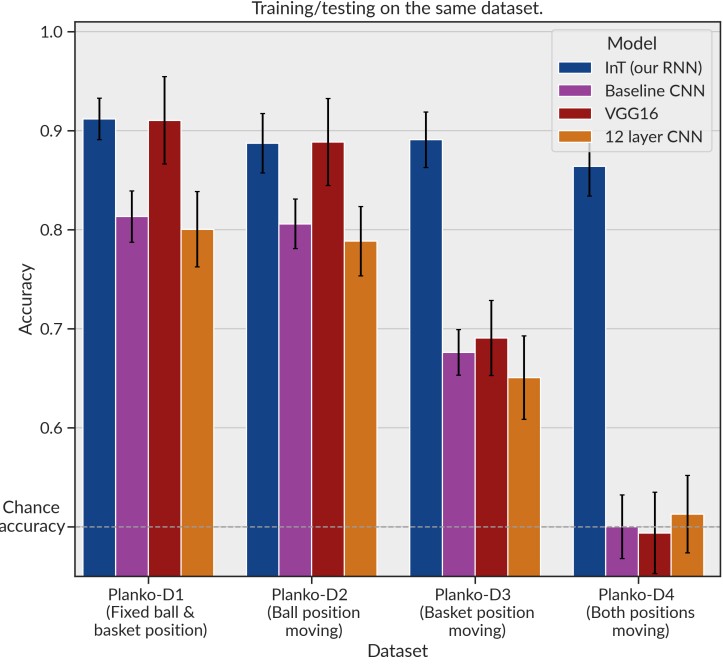

Figure 2: **FFN and RNN performance on the *Planko* challenge.** Error bars depict 95% bootstrapped confidence intervals. The InT is significantly more accurate than any other model on the most challenging versions of *Planko*: when the basket position or both the basket and ball positions are randomly placed across stimuli.

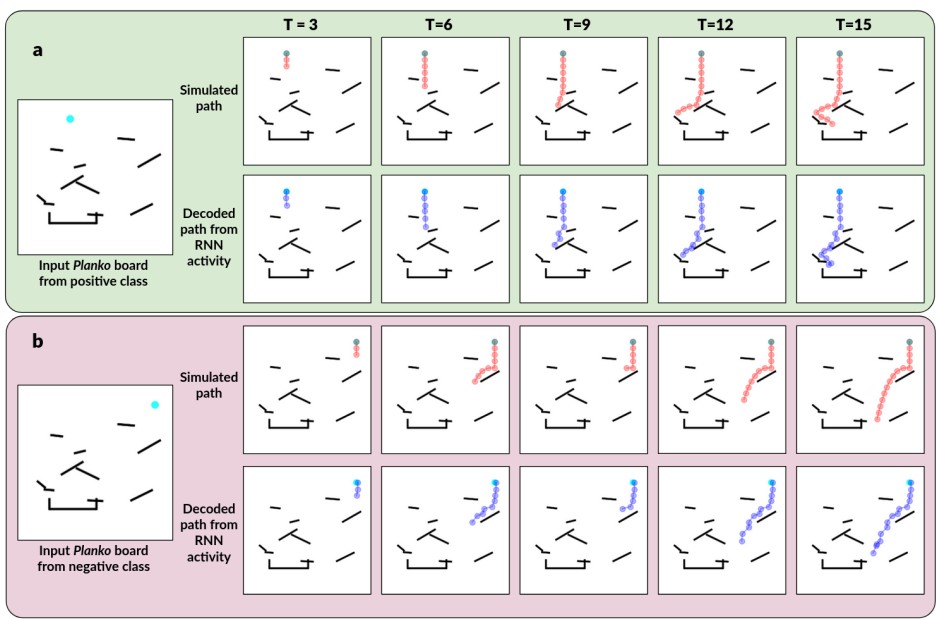

Figure 3: **The InT learns to solve *Planko* by learning a visual simulation strategy that resembles the ground truth physics.** Sample positive and negative *Planko* boards were shown to an InT while decoding the position of the ball from the activities of excitatory units. Ground truth paths are depicted in red and decoded paths from the InT are depicted in blue.

it falls downward is tracked to determine if it eventually lands in the catcher (positive class) or falls to the ground (negative class). Figure 1 illustrates example *Planko* ball trajectories for both positive and negative classes.

## 2.1 Parameterizing *Planko* board difficulty

By bounding the variations in the *Planko* board elements, we systematically control for the challenge associated with solving the board for neural network models. *Planko*-D1 (Figure. 1a) is the easiest variant of the task in which both the ball and catcher positions are constant across the entire dataset. *Planko*-D2 (Figure. 1b) and *Planko*-D3 (Figure. 1c) are intermediate-level boards. While in *Planko*-D2 the initial ball position is randomly sampled from the upper $40\%$ of the game board (with the catcher position remains constant), *Planko*-D3 places the catcher in a random location sampled from the lower $40\%$ of the game board (with the ball position constant). *Planko*-D4 (Figure. 1d) is the version of the task wherein both the ball and catcher positions are stochastic. Boards in which the ball hits either vertical wall are excluded from the data used for the neural network analysis.

## 3 RNNs, but not FFNs, solve *Planko*-D4

**General setup** All models used herein were trained to classify each *Planko* game board into one of either positive or negative classes. Model parameters were optimized with Stochastic Gradient Descent implemented via the Adam algorithm Kingma & Ba (2014) with an initial learning rate of $3e-4$. Binary Cross Entropy (BCE) was used as the training objective. Each train (test) dataset consisted of $200K$ ($5K$) *Planko* boards of dimensions $64 \times 64$ pixels. Training was carried out on a NVIDIA TITAN Xp GPU for 100 epochs while measuring validation accuracy after each epoch over a held-out set of $10K$ boards.

**The InT Model** The Index-and-Track circuit (a complete model description in Linsley et al. 2021) architecture consisted of an input layer with $64$ $1 \times 1$ convolutional filters followed by the InT circuit with $3 \times 3$ horizontal kernels and $64$ output channels. A $1 \times 1$ convolutional "readout" followed by a linear layer transformed the final RNN hidden state to the classification output. The RNN is trained for $T = 24$ time steps.

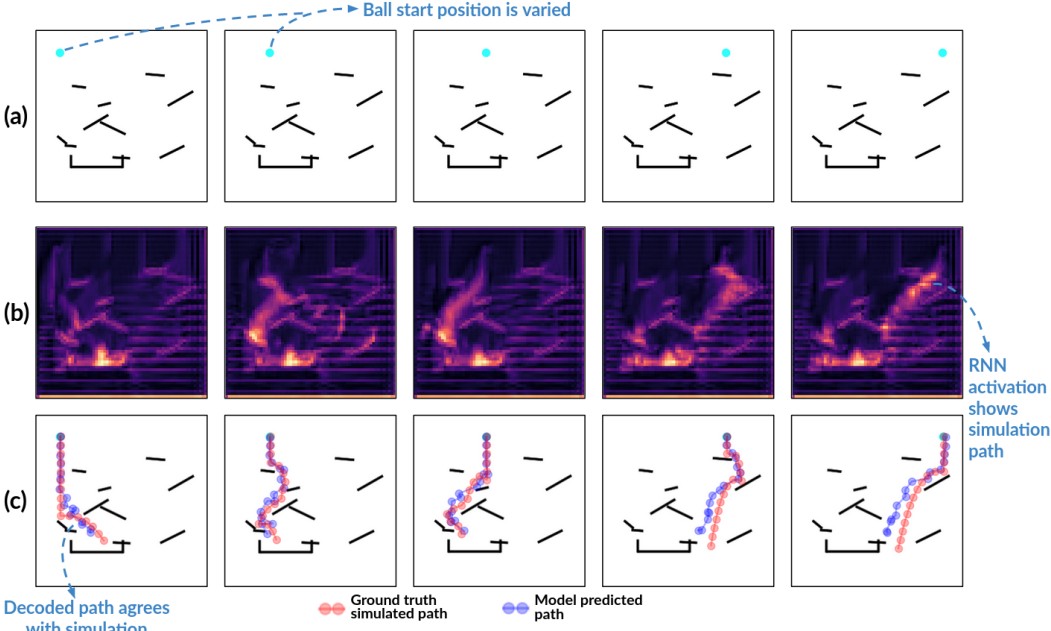

Figure 4: **The InT's attention maps reveal solution strategies.** (a) Testing the RNN (InT) with boards where everything is held constant except the ball which is horizontally translated. (b) The hidden state activity at timestep T for the RNN for the boards in (a). (c) The decoded ball positions from the hidden state vs the ground truth simulated paths.

**Baselines** In addition to the InT, we trained a simple feedforward 2-layer convolutional neural network (termed "Baseline CNN"), a standard VGG16 Simonyan & Zisserman (2014), and a 12-layer CNN with a parameter count identical to the InT circuit.

**Classification results** The performance landscape of models across the *Planko* tasks revealed that both FFNs and RNNs solved easier versions of the task (Figure. 2). However, the InT was significantly more accurate on *Planko*-D4, the most challenging version.

**Decoding analysis** We train a decoder to extract the coordinates of the *Planko* ball positions from the final timestep InT activities. The decoder is trained to minimize the mean-squared error between the predicted ball coordinates and the ground truth ball position obtained from the physics simulator. The decoder consisted of three layers of $1 \times 1$ convolution and pooling operations and finally a linear readout layer. The model was trained on $64$ channel $64 \times 64$ feature tensors from the final timestep of the trained InT ciruit. A total of 16 decoder models are trained for each of the 16 ball positions from the simulator for 20 epochs with 200,000 feature tensors. The mean-squared error is measured on the validation set after every epoch and the model with the least error is used to predict the ball position from new boards.

# 4   Conclusion

We explore the conditions under which "visual simulation" emerges as the naturally optimal algorithm in task-optimized RNNs. We demonstrate that only the most performant RNN, on our most variable task, adopts a "simulation" strategy. To the best of our knowledge, we provide the first evidence that information about imaginary future states can be decoded from RNN internal representations. While this work is preliminary, we are hopeful that it paves the way for RNN-guided electrophysiology research to understand the mechanistic basis of visual simulation.

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
