# OpenReview forum: "The emergence of visual simulation in task-optimized recurrent neural networks"
_NeurIPS.cc/2022/Workshop/SVRHM — SVRHM Poster_

### Official Review · Reviewer_ZMCk · 2022-10-14
**Great study! (results have to be better contextualized per existing literature)**

**Rating:** 8
**Confidence:** 4

**Review:**

In this paper, the authors use the Planko task to show that simulation-like representations arise in deep neural networks trained to solve the task only when the task variability is high. This work shows that visual simulation may be a task-specific optimal strategy and provides a computational framework to understand the process. This is a simple yet solid work that advances our understanding of the computations underlying forward simulation and elucidates when such a strategy might be useful for planning effective behavior.

Comments:
1. When it comes to physical scene understanding, there are two contrasting viewpoints in the literature: 1) future prediction is done through pattern recognition in the ventral visual pathway and as implemented in feedforward CNNs; 2) prediction of future states of the scene is done through forward simulation of what will happen next (as implemented in video game physics engines). The authors have set up a nice task to show that both strategies can potentially help depending on the task context (and difficulty): VGG-16 being the top model in Planko-D1 and Planko-D2; and the InT model rising above VGG-16 only in Planko-D3 and Planko-D4. I suggest the authors to interpret their findings in the context of this ongoing debate.

References:

[pattern recognition viewpoint]
a. Lerer, A., Gross, S. & Fergus, R. Learning physical intuition of block towers by example. 33rd Int. Conf. Mach. Learn. ICML 2016 1, 648–656 (2016).
b. Conwell, C., Doshi, F. & Alvarez, G. Human-Like Judgments of Stability Emerge from Purely Perceptual Features: Evidence from Supervised and Unsupervised Deep Neural Networks. in 2019 Conference on Cognitive Computational Neuroscience 605–608 (Cognitive Computational Neuroscience, 2019). doi:10.32470/CCN.2019.1406-0

[forward simulation: in addition to the ones already cited]
c. Zhang, R., Wu, J., Zhang, C., Freeman, W. T. & Tenenbaum, J. B. A Comparative Evaluation of Approximate Probabilistic Simulation and Deep Neural Networks as Accounts of Human Physical Scene Understanding. in Proceedings of the 38th Annual Meeting of the Cognitive Science Society 1–6 (2016).

2. The authors claim that: “Our findings indicate that robust visual simulation emerges as an optimal algorithm in difficult environments, and that prior work Rajalingham et al. (2021, 2022) suggesting that additional learning constraints are needed for visual simulation may be a byproduct of a trivial task driving models to learn shortcuts”. However, it is not clear how the current study shows that the additional constraints used in Rajalingham et al are byproducts of a trivial task. In Rajalingham et al, the monkeys were trained to play pong even when there was an occluder on the screen thereby increasing the task difficulty and potentially necessitating some form of simulation to keep track of the ball position behind the occluder. They also show that an RNN trained without the ‘simulation’ constraint cannot solve the task on such occluded trials. So, the authors should clearly explain how their claim relates to these studies both in terms of task constraints and computational modeling.

---

### Official Review · Reviewer_1M6S · 2022-10-16
**The emergence of visual simulation in task-optimized recurrent neural networks**

**Rating:** 5
**Confidence:** 4

**Review:**

First of all, I believe the topic discussed in the present manuscript is novel, timely, and interesting. I find the overall framework proposed here to be of high relevance and interest to the audience of the SVRHM workshop. With that said, I’m not entirely convince that the main argument/conclusion can actually be drawn from the results/evidence presented in the manuscript. Below, I included specific comments related to this point.
1.	In the Planko challenge, the authors ‘controlled for variations in the task space to explicitly prevent the learning of shortcut solutions’. It is not clear from a biological perspective why this manipulation is important. To optimize task performance or to generate adaptive behavior in general, wouldn’t an organism want to achieve the opposite i.e., search the solution space for shortcuts or most efficient solutions? It would be helpful if the authors could explicit discuss the value/significance of this essential feature of the task design on the get-on so the readers could appreciate and properly interpret the reported findings in the manuscript.
2.	In Section 3, I’m not sure it is completely fair to say that FFNs cannot solve Planko-D4. I’m curious if there are specific configurations/architectures within the FFN-scheme that can be added to the models that would actually allow the models to sufficiently solve Planko-D4. It would be helpful if the authors could discuss these possibilities, especially in the context of biology.
3.	In the conclusion, I think it would be helpful if the authors could tie the results (in light of the specific task design and manipulations e.g., trial variability, difficulty) back to past literature and what’s currently known both empirically and theoretically in the space of ‘visual simulation. My impression is that this visual simulation only emerges in a very specific task scenario which works against the idea that visual simulation is an adaptive process shaped through evolution to optimize behavior.

---

### Official Review · Reviewer_GpZP · 2022-10-17
**Promising results and interesting experiments**

**Rating:** 6
**Confidence:** 4

**Review:**

In this paper, the authors have explored the emergence of visual simulation as a meaningful computational strategy in solving tasks, similar to the behavioral mechanisms that take place in primates - in this case, the Planko task. With tasks that have their own individual constraints, it's important to see if simulation is an emergent property that underlies the decision processes of DCNNs. In this submission, experiments were performed with the usage of feed-forward, recurrent, and attention models on varying difficulty levels of the task. The authors were able to show that an attentional circuit model from (InT, Linsley et al. 2021) had visual simulation strategies that mimicked that of ground physics.

The authors showed that the internal representations from RNNs seemed to present knowledge about how the task environment would look like in the future states. Such a finding could be really informative for the future work of mental modelling and predictive coding.

This reviewer consider this paper to be revealing of the underlying computational mechanisms that take place within DCNNs as they perform fundamental predictive tasks. However, this kind of idea has been used in other publications where the usage of recurrence is used as a technique in solving hard tasks (such as mazes, or paths). It would be useful to provide some numerical insight to show the differences in varying accuracies between difficulty levels of the Planko task among the models as well.

---

### Official Review · Reviewer_GAJ3 · 2022-10-17
**Intriguing task and model analysis, but unclear on analysis details**

**Rating:** 7
**Confidence:** 3

**Review:**

The authors investigate the emergence of visual simulation in neural networks by extending a previously developed task, in particular using multiple grades of difficulty. They find that a neurobiologically inspired model results in better performance than a parameter-matched convolutional neural network and their model analysis suggests the emergence of visual simulation in that network. I thought the task was well-suited to analyzing neural networks and the decoding analysis was suggestive of an intriguing emergence. However, this analysis was barely explained in the manuscript and I had to fill in a lot of gaps with my own best guesses. In particular, the text doesn't reference the latter two figures at all. In light of this lack of explanation, it is hard for me to judge the rigor of the decoding analysis and I would emphasize the importance of clarifying this analysis. Nevertheless, this submission is a promising attempt at investigating visual simulation and I recommend acceptance. I'm curious to see where this work will lead.